# Microbiota Ecosystem in Recurrent Cystitis and the Immunological Microenvironment of Urothelium

**DOI:** 10.3390/healthcare11040525

**Published:** 2023-02-10

**Authors:** Mattia Dominoni, Annachiara Licia Scatigno, Marco La Verde, Stefano Bogliolo, Chiara Melito, Andrea Gritti, Marianna Francesca Pasquali, Marco Torella, Barbara Gardella

**Affiliations:** 1Department of Clinical, Surgical, Diagnostic and Paediatric Sciences, University of Pavia, 27100 Pavia, Italy; 2Department of Obstetrics and Gynecology, IRCCS Foundation Policlinico San Matteo, 27100 Pavia, Italy; 3Department of Woman, Child and General and Specialized Surgery, Obstetrics and Gynecology Unit, University of Campania “Luigi Vanvitelli”, 81100 Naples, Italy; 4Department of Obstetrics and Gynecological Oncology, P.O del Tigullio” Hospital-ASL4, Metropolitan Area of Genoa, 16034 Genoa, Italy

**Keywords:** recurrent urinary tract infection, vaginal microbiota, urinary microbiota, dysbiosis

## Abstract

Urinary tract infections (UTIs) represent one of the most frequent low genital tract diseases in the female population. When UTIs occur with a frequency of at least three times per year or two times in the last six month, we speak of recurrent UTI (rUTI) and up to 70% of women will have rUTI within 1 year. It was previously thought that antibiotic resistance was principally responsible for the recurrence of UTIs, but nowadays new diagnostic technologies have shown the role of microbiota in the pathophysiology of these diseases. Much research has been conducted on the role of gut microbiome in the development of rUTI, while little is known yet about vaginal and urinary microbiome and the possible immunological and microscopical mechanisms through which they trigger symptoms. New discoveries and clinical perspectives are arising, and they all agree that a personalized, multi-modal approach, treating vaginal and urinary dysbiosis, may reduce rUTIs more successfully.

## 1. Introduction

A total of 1.5 million people suffers from urinary tract infections (UTIs) every year, making it one of the most prevalent health problems [1]. Women experience UTIs eight times more often than men [2], and 50–60% of adult women will have at least one UTI in their lifetime, affecting their quality of life and psychological wellbeing [3,4].

Anatomical characteristics, sexual behaviour, urogenital aging, pelvic organ prolapse, urethral diverticula, vescico-vaginal fistula, urinary incontinence, menopause, and pregnancy all represent possible risk factors for women [5]. 

In clinical practice it may be useful to distinguish between uncomplicated and complicated UTIs. Complicated UTIs are caused by urological anomalies, including indwelling catheters, renal insufficiency, neurogenic bladder, pregnancy, previous urological surgery, and conditions causing an immunocompromised state [6], and these may also progress to sepsis and other systemic illnesses, which mostly impact the kidneys [6]. 

Recurrent UTIs (rUTIs) are characterised as complicated and/or uncomplicated UTIs that happen at least three times yearly or twice over six months [7,8], differently from persistent infections in which the pathogen is not eradicated but instead persists in some of the infected people’s cells [9]. Recurrent UTIs are common; after getting one, 24% of women will get another within 6 months, and up to 70% will get another within a year [10,11]. Six or more episodes of rUTIs occur in at least 35 million women worldwide each year (1% of all women) [10,11,12].

The pathophysiology of rUTIs is not well understood. However, in the 80s, it was already clear that recurrence of UTIs was closely linked to antibiotic resistance [13,14,15]. The increased use of antibiotics globally, along with prophylactic therapy, contributed to the development of multiresistant bacteria, such as extended-spectrum beta-lactamase-producing bacteria, carbapenemase-resistant organisms, and pan-resistant bacteria [16,17]. 

Furthermore, the myth that urine is sterile has been dispelled only in recent years by developments in technology and molecular biology, concomitantly with the discovery of the role of microbiota of the bladder, vagina, and gut in the pathophysiology of rUTIs [18,19,20]. While little is known about the vaginal and urinary microbiomes, a great deal of study has been done on the function of the gut microbiome in the development of rUTIs [21]. 

The purpose of this review is to highlight potential mechanisms by which the vaginal and urinary microbiomes, as well as the potential role of the urothelial immunological microenvironment, contribute to rUTIs onset in women of different ages (Figure 1). 

## 2. Materials and Methods

The most significant medical databases, including PubMed, Cochrane Database of Systematic Reviews, EMBASE, and Web of Science, were consulted, according to a combination of the following keywords: “recurrent urinary tract infection, recurrent cystitis, vaginal microbiome, vaginal microbiota, urinary microbiota, urinary microbiome, urobiome, dysbiosis, urinary bladder disease”, including pluralization and English spelling variations and suffixes/prefixes. From 2000 until 11 November 2022, we collected all publications, including case studies, literature reviews, and prospective or retrospective trials. Two authors (MD and ALS) independently evaluated the references to incorporate the literature data into the review. Preferred reporting items for systematic reviews and meta-analyses (PRISMA) method was applied to conduct a systematic search (Figure 2).

In the first step, the authors considered the title of the paper, then the abstract, and finally the manuscript. Consequently, the data obtained was collected. Studies were considered qualified if they met the following criteria: (I) the involvement of the vaginal microbiota and microbiome in the onset of rUTIs in female population, (II) the role of the urinary microbiota and microbiome in rUTIs in women, (III) dysbiosis as a cause of recurrent cystitis, and (IV) novel therapeutics approaches in the field of study. Instead, the following were considered exclusion criteria: (I) case reports; (II) conference abstracts, editorials, and pre-prints manuscripts; (III) multimedia; and (IV) papers written in languages other than English.

To ensure validity and prevent any selection, performance, detection, attrition, and reporting bias, two researchers (MD and ALS) independently assessed the risk of bias for each selected study, in accordance with the Cochrane Handbook for Systematic Reviews of Interventions [22,23]. Conflicts were resolved through discussion between researchers. Finally, the two researchers examined and extracted data separately. 

## 3. Results

The search method provided 201 papers in total and another 10 studies were included through the references. In total, 90 publications were screened by title and/or abstract following the elimination of articles not published in English or published before 2000. Duplicate papers, and irrelevant works were excused from the analysis. In the end, 31 research articles were included and analysed (Figure 1). Table 1 reports the main findings derived from the literature data about the internship between microbiota, immunology, and rUTIs.

### 3.1. Vaginal Microbiome and rUTIs

The pathogenesis of rUTIs is significantly influenced by the vaginal microenvironment, in contrast to widespread assumption, which attested that bacteria causing UTIs typically originate from the gut altered microbiota, as the only way of infection [24]. 

Community state types (CSTs) represents a classification used to describe at least five major subcategories of cervico-vaginal bacterial species involved in the maintenance of vaginal balancing between physiological and pathological flora. Each group has a unique mix of bacteria, each with a different relative characteristic. Four of these are dominated by one of the following four species: *L. crispatus* (CST I), *L. gasseri* (CST II), *L. iners* (CST III), or *L. jensenii* (CST V) [25,26]. 

Instead, CST IV predominantly contains anaerobic bacteria, such *G. vaginalis, Atopobium vaginae*, and *Megasphaera spp*., similar to the vaginal microbiota in bacterial vaginosis [25,26]. Interestingly, a recent meta-analysis reported that *Prevotella bivia*, *G. vaginalis*, *Chlamydia trachomatis*, and *Human Papillomavirus* infections are more common in women with lower levels of *Lactobacillus* in their CST IV cervico-vaginal microbiota than in women with higher levels of *Lactobacillus* [27]. 

Several *Lactobacillus* species, such as *L. crispatus*, *L. jensenii*, *L. gasseri*, and *L. iners,* constitute most of the vaginal microbiota in women of reproductive age [28,29,30]. These bacteria produce lactic acid, helping the maintenance of the vaginal acidic pH [31,32]. 

By creating bacteriocins and hydrogen peroxide, vaginal lactobacilli perform a protective function, inhibiting the colonisation of other potential pathogens, particularly *E. coli* [28,33,34,35,36]. For this reason, lower lactobacilli levels promote the insurgence of bacterial vaginosis or vaginal *E. coli* colonisation, which increase the risk of UTI insurgence [30,37,38,39].

The vagina also represents a reservoir for pathogens: The literature data underlines that women with a history of UTI exhibit more *E. coli* colonisation in the vaginal introitus (>105 CFU/mL), highlighting the importance of vaginal microenvironment in the pathogenesis of rUTIs [40].

*S. saprophyticus* is the most frequent gram-positive bacterial source of community-acquired UTI. *S. saprophyticus* virulence in in vitro and rat UTI model is based on the following components: Secreted surface-associated proteins Aas (hemagglutinin) and Ssp (lipase), the proteins UafA of cell wall, SdrI, SssF, and UafB, which mediate adherence, and the ureases [30].

By contrast, *S. aureus* and *S. epidermidis* can also cause UTI, especially during catheterization or pregnancy: Experimental models highlighted that the nickel ABC-transporters Opp2 and Opp5a are involved in the pathogenesis of *S. aureus* urinary infection [30,41,42,43,44]. 

Finally, vaginal bacteria, such as *Actinobacteria*, other *Firmicutes*, and gram-negative anaerobic organisms, which are not common uropathogens, may colonise the urinary system, alter the physiological microbiota, and change the immunological assessment of vaginal and bladder mucosa. In other words, even if specific vaginal bacteria do not colonise the bladder or are eliminated by the host prior to the diagnosis of a UTI, brief contact with these bacteria in the urinary tract can still have a significant impact on UTI pathogenesis [30]. This phenomenon is called “covert pathogenesis” [30,45]. For instance, group *B streptococcus* and *G. vaginalis* promote the survival of *E. coli* in the bladder, permitting the development of UTIs [46,47,48]. Numerous studies have connected *Streptococcus agalactiae* (GBS) colonisation to vulvo-vaginitis [49] and urinary tract infections [50], but none have looked at the connection between vaginal GBS and GBS UTI [30]. GBS colonisation is typically asymptomatic. By contrast, *Gardnerella vaginalis* can also cause UTI and is connected to sepsis, renal disease, and urgency incontinence [30,51], as we better described later (Section 4.2).

### 3.2. Urinary Microbiome and rUTIs

Modern urine culture techniques have shown that several bacteria allow the maintenance of urothelium homeostasis [52,53,54]. The host characteristics, which change across people, life, and geographical areas, as well as environmental exposure and behavioural factors, are the most important factors for maintaining the balance of this microbial ecosystem [55].

The most often identified colonising microorganisms in the urine microbiome are *Lactobacillus* and *Streptococcus*, which constitute a barrier against infections, producing factors, which inhibit the adhesion of pathogens to epithelium, such as lactic acid. *Alloscardovia*, *Burkholderia*, *Jonquetella*, *Klebsiella*, *Saccharofermentans*, *Rhodanobacter*, and *Veillonella* are less often identified bacteria [56]. *Proteobacteria* (35.6%), *Firmicutes* (31.3%), *Actinobacteria* (22.4%), *Bacteroidetes* (6.4%), and others (4.3%) are the key phyla of the human urinary tract, according to Morand et al. [57]. The authors attested that the urine microbiota ordinarily contains several uropathogens, but pathogenicity results from an imbalance in their relative percentages and from the host immunological response [58]. 

From 435 urine samples, Dubourg et al. identified 450 different bacterial species, of which 256 had never been discovered before in urine, while 18 were entirely new [58].

Other recent research suggests that many urinary system diseases, including UTIs and rUTIs, may be influenced by the urine microbiome (or urobiome), which plays a key role in the maintenance of the homeostasis of urothelial microenvironment [59]. Urinary microbiota abnormalities precede UTI onset, and the urobiome normalizes following therapy, as Bossa et al. demonstrated [60].

The knowledge of the urobiome is fundamental in clinical practice because different clinical manifestations are probably connected to a specific urine microbiota modification, as attested by Burnett et al. [61]. Additionally, non-infectious urologic conditions, such as neurogenic bladder dysfunction, interstitial cystitis, and urgency urine incontinence have been associated to changes in the urinary microbiota spectrum [54]. 

It is interesting to consider in the knowledge of a UTI’s pathology and in the importance of health urobiome is the evidence that uropathogenic Escherichia coli (UPEC) has a reservoir also in the bladder epithelium; many investigations in both adults and children, as well as in bladder biopsies, demonstrated intracellular UPEC in bladder epithelial cells that release in urine [62,63,64,65,66]. In addition, 82% of rUTIs are brought on by the same UPEC strain as in the prior infection, even when the proper antibiotic therapy is administered [24,67,68,69,70,71]. The pathogenesis of UPEC is described in Section 4.2.

## 4. Discussion and Conclusions

### 4.1. Risk Factors for rUTIs in Women

Hormonal fluctuations have a significant role in the changes in vaginal and urine microbiomes composition; oestrogen promotes *Lactobacillus* development in the bladder and vagina, increasing their defensive role against pathogens and infections. Consequently, loss of oestrogen in postmenopausal women causes a reduction in vaginal lactobacilli and an increase in rUTIs [72,73]. The genitourinary syndrome of menopause is constituted by vaginal epithelium thinning, a decrease in extracellular matrix, proteoglycans and collagens synthesis, and vulvovaginal atrophy [74], which facilitate the penetration of bacteria in urothelium and vaginal epithelium. 

For this reason, post-menopausal women are more prone to develop rUTIs, with a rate of 8–11% [75,76,77,78]. Regarding vaginal and urinary microbiome composition after menopause, some authors demonstrated that post-menopausal women present fewer distinct bacterial species [79,80,81], while others supported the development of an increased diversity of species [82]. 

UTIs are also a common problem among pregnant women, representing the most common infection during this period of life, particularly asymptomatic bacteriuria, affecting 2 to 7% of pregnant women [83]. As we all know, pregnancy is characterized by physiological changes in immune response, the vaginal microbiome also undergoes major changes. Indeed, pregnancy reduces the differences in microbiome diversity across women; particularly, pregnancy-related vaginal alterations result in an increased *Lactobacillus* dominance and a reduced species diversity [84]. These changes are protective regarding a preterm birth rate because they boost infection resistance and support the production of anti-inflammatory cytokines [85]. Furthermore, they regard also racioethnic differences in the vaginal microbiome, so particularly in women of African ancestry, the configuration of the vaginal microbiome during pregnancy may have predictive value for premature birth [86].

Other factors that influence vaginal microbiome composition are:Contraceptive methods: Spermicidal products containing nonoxynol-9 deplete lactobacilli and favor *E. coli* colonization [87,88,89,90,91,92]. Instead, oral contraceptives seem to decrease the rate of bacterial vaginosis [93], but they do not influence the risk of rUTIs;Sexual activity: Vaginal activity either promotes the entry of possible germs into the urethral meatus from the vagina or facilitates the transfer of potential uropathogens to the vagina [10,48,94,95,96,97,98];New antimicrobial treatments (oral or topical) for the risk of the development of antibiotic resistance [16,17].


### 4.2. NGS as a Better Diagnostic Tool

Nowadays, several studies and research articles have re-written the idea that most UTI bacteria originate in the gut [99], and recent research has clarified the role that urine and vaginal bacteria play in the onset and recurrence of these diseases [100,101]. The microbiomes of the vagina and urinary tract are inextricably related and together participate in the maintenance of a healthy balance in the genital and urinary tracts [19]. In addition, from a microbiological perspective, around one-third of the bladder microbiota only resides in the vagina [102]. Mechanical transfer is one of the main risk factors for UTIs and rUTIs [10,48,94,95,96,97,98,103] because it enables vaginal bacteria to enter the urinary system, such as during sexual activity [33,70].

In this new perspective, the myth that urine is sterile was disproved also thanks to the development of new analytical techniques, such as NGS and metagenomic approaches [18], which allowed for the detection of a microbiome in the healthy urogenital tract [104,105,106]. Microbial ecologists created the culture-independent DNA-based identification of microorganisms with the aim of identifying bacterial species without the need for culture. Particularly, NGS employs PCR amplification and high-throughput sequencing of essential 16S rRNA genes, using polymorphisms of the 16S rRNA gene amplicon to distinguish bacterial species, even those that are closely related [19,107]. A urine sample is sequenced using a multi-step process that starts with the isolation and purification of microbial DNA, follows with 16S rRNA amplification and sequencing, and ends with bioinformatic analysis through a variety of software database platforms. As a result, there are still a lot of restrictions with this technology, particularly in terms of its clinical uses [18].

Yoo et al. demonstrated that the clinical application of urine NGS in cases of acute uncomplicated cystitis and rUTIs reported a better sensitivity than the application of conventional urine culture [82], which is consistent with prior research [108,109,110,111]. Indeed, it appears that a typical urine culture misses roughly 90% of non-UPEC pathogens [110], and anaerobic bacteria or a multi-microbial illness may be to blame for negative findings in routine urine cultures [107,112,113]. 

Most importantly, NGS is not greatly impacted by antibiotic usage, because bacteria do not need to be alive as for a traditional culture method [114]. Furthermore, NGS is highly sensitive to atypical bacteria, anaerobes, or multimicrobial urinary tract infections [113]. Another crucial element that facilitates prompt clinical decision-making and medication is represented by the faster NGS technique for the detection of pathogens with respect to culture; this reduces testing times from several days to just 24 h [114].

### 4.3. Pathophysiology and Immunology in rUTIs

While analysing the immunological assessment of urinary infections, data in the literature reported that intracellular bacterial communities (IBCs) and quiescent intracellular reservoirs (QIRs) are two methods that allow pathogens to survive antibiotic treatment and to host an immune response in the bladder, developing a chronic colonization [63,66].

Regarding UPEC, adhesive organelles, such as type 1, P, S, and F1C pili, are used to first infiltrate the host cells in the urothelium. Then, UPEC creates IBCs, which consist of the development of a biofilm formed of a polysaccharide matrix wrapped in a uroplakin coating, enabling UPEC to proliferate and thrive in a secure manner [64,115,116,117,118,119]. As opposed to this, QIRs are made up of a subgroup of bacteria that have remained undetected by the host immune system for a considerable amount of time in cells after receiving antibiotics [120,121]. Dormant bacteria may begin to reproduce and lead to reinfection because of the urothelium’s turnover. IBCs are transitory, developing within a few hours in the cytosol as opposed to QIRs, which might spend months quiescent within the endosomes [122]. 

Lipopolysaccharide (LPS), a key component of UPEC pathogenicity, affects UPEC life cycles and promotes reservoir development [123] by activating intracellular signalling pathways and innate and adaptive immune responses [124]. By raising cytosolic calcium through a Toll-like receptor (TRL 4-mediated increase), LPS suppresses the synthesis of cytokines [125]. Additionally, NLRP3 inflammasome activation by pathogen-associated molecules, such as flagellin and hemolysin as well as LPS can cause urothelial cells to exfoliate and let UPEC to enter deeply [126,127].

Regarding the relationship between the innate and adaptive immune systems and rUTIs, this is not completely understood yet [122].

The functions of pentraxin 3 (PTX3) and uroplakin IIIa (UPIIIa) signalling have received little attention in the literature. A crucial function of PTX3 is that polymorphisms or a deficit in it may impair the body’s capacity to control infection, which may promote infection spread [128]. The endocytic process is instead induced by UPIIIa signalling, which enters the intracellular space [129].

Instead, particular attention has been paid to the relation between vaginal microenvironment and urinary tract inflammatory diseases. The assumption that the vagina is the main source of bladder colonising pathogens was made since women have UTIs at a greater rate than males [115,116]. The vaginal canal can operate as a reservoir for *E. coli* and other bacteria, becoming a significant player in the pathogenesis of UTI. 

*E. coli* can penetrate vaginal cells and remain in the vagina during UTI, according to preliminary research in murine UTI models [117]. Regarding this, more research has been conducted on the relationship between *Gardenerella vaginalis* and *E. coli.*

Animal experiments that exposed the urinary system to different common vaginal bacteria (especially *Gardnerella*) in the setting of *E. coli* UTI corroborate the previous reported theory of “covert pathogenesis” [30,71]. 

*Gardnerella* can frequently be found in urine samples from healthy, asymptomatic women. Three patterns of patients who tested positive for *Gardnerella* were proposed by Yoo et al. [71]: (I) the *Escherichia*-dominant group; (II) the *Gardnerella*-dominant group; and (III) the *Lactobacillus*-dominant group. They emphasised that all *Escherichia* dominant groups were linked to rUTI, but *Gardnerella-* and *Lactobacillus*-dominant groups might be linked to rUTI but not necessarily be symptomatic. This supported the idea that bladder dysbiosis can cause various symptoms by altering the immune system’s reaction to bacterial colonisation [71]. Furthermore, it was shown that *Gardnerella* may be a “covert” pathogen that causes *E. coli* activation [69], and UTI can also happen in the *Lactobacillus*-dominant group even if a minor amount of *Gardnerella* is present, if *Lactobacillus* has a poor protective effect [71]. 

Other research confirmed that the development of UPEC from bladder reservoirs is significantly influenced by *Gardnerella*. Indeed, it influences urothelial apoptosis and exfoliation and other mucosal immune system-related activities as demonstrated in a mouse model [130,131]. Among these, immediate-early (IE) genes, including the orphan nuclear receptor Nur77 (also known as Nr4a1), are increased in mice exposed to *Gardnerella* [131,132]; at the same time, animals lacking Nur77 are not at risk from recurrent UPEC UTI after *Gardnerella* exposure. 

Numerous cellular functions are controlled by Nur77, including apoptosis in various tissues [133,134]. Additionally, Nur77 regulates inflammation [135] and has a specific impact on T-cell responses [136] and Ly6C-monocytes [137]. As a result, the IE response could play a role in the *Gardnerella*-related recurrent UPEC UTI [138]. 

Instead, KEGG pathways and GO keywords are significantly changed after several *Gardnerella* exposures [138]. IL-12, IFN-g, and RANTES levels rise in bladder homogenates after exposure to *Gardnerella* [48], and pathways associated to T and B cells are also activated.

Finally, Kirjavainen et al. investigated immune defence anomalies in women with rUTIs and discovered that peripheral monocytes and myeloid dendritic cells (DCs) produced elevated level of interleukin-12 and did not induce the T cell activation. In the case of rUTIs, the T cell polarisation is avoided. In addition, there was a decrease in levels of vascular endothelial growth factor (VEGF) related with tissue healing and a reduction in concentrations of monocyte chemotactic protein 1, the main chemoattractant for DC and monocytes [139]. 

All these factors may promote the insurgence of urinary infection and chronic colonisation due to the deficiency of immune response and the imbalance of host response to bacterial injuries. It is likely that the host immune response depends on the phenotypic and behaviour characteristics (such as smoking, sexual activity, alcohol abuse, and menopausal status) of a host as previous described. 

### 4.4. New Perspectives of Therapy and Prevention

Recurrent UTIs are strictly associated with urinary tract dysbiosis [18,104]. The importance of lactobacilli and oestrogens in the prevention of rUTIs was confirmed by Neugent et al., who described the correlations between *Bifidobacterium*, *Lactobacillus,* and urinary oestrogens in women with no history of UTIs [140]. According to this theory, some researchers suggested that administering probiotics may be more beneficial in treating rUTIs [141,142] as opposed to administering antibiotics or taking antibiotics prophylactically at low doses to prevent recurrent infections, both of which promote the evolution of pathogenic resistance by causing bacterial persister cells [143]. 

A considerable decrease in rUTIs is linked to the use of Lactobacillus vaginal suppositories [38,55,144]. Sadahira et al. showed that the administration GAI 98,322 strain of *L. crispatus* had a significant effect in reducing the recurrent cystitis in 86% of patients. However, more importantly, the suppressive effect persisted in 77% of patients for at least a year after the end of the therapy, with a significant decrease in the mean number of cystitis episodes both during and after administration [145]. The oral treatment with *Lactobacillus reuteri RC-14* and *Lactobacillus rhamnosus GR-1* also improved the population profiles of vaginal lactobacilli and reduced the colonisation of potentially dangerous bacteria [146].

In order to support the host’s immunological assessment against bacterial invasion and prevent recurrent infection, functional restoration should be the main focus of therapy, according to the recent literature data on the urinary tract urobiome and the importance the local and systemic immune system response in the prevention of UTIs recurrence [147].

As was already mentioned, oestrogen regulates the balance of the urogenital microbiome; it promotes lactobacilli growth, whereas oestrogen insufficiency results in a decrease in vaginal lactobacilli, which raises the risk of rUTIs [72,73]. Therefore, rUTIs may be decreased by oestrogen replacement treatment [73,94,148,149,150,151], and intravaginal oestrogen may provide great benefit with less risk when compared to oral oestrogen [152]. 

Recently, research has been conducted on the use of natural sources for therapy and prevention of rUTIs. For example, Mehta et al. studied the potential antibacterial role of the oroxindin from *Bacopa monnieri* against UTIs caused by *Klebsiella pneumoniae* and *Proteuns mirabilis. B. monnieri* is a medicinal plant growing in the world’s wetlands and warmer regions; the authors demonstrated that *K. pneumoniae* and *P. mirabilis* can be effectively eliminated by *B. monnieri*, also establishing its safety [153]. 

## 5. Conclusions

In conclusion the complex correlation among microbiota, low genital tract, and urinary system is based on the balance between host characteristics, immunological microenvironment and pathogens. Further investigation may provide an accurate analysis of the urogenital microbiome, especially to promote a tailoring therapy in order to reduce antibiotic resistance and increase the physiological mechanism of urothelium response.

**Table 1 healthcare-11-00525-t001:** Main findings from the studies included in the review.

Variables	Main Findings
Role of vaginal microbiome [30,37,44,59,71,72,101,131,138,142]	The vaginal microbiome is involved in rUTIs pathogenesis: if its balance is maintained, it constitutes a barrier against pathogens. However, every change, which we know as bacterial vaginosis, is an important risk factor for the development of urinary tract infections.
	This may be a consequence of the decrease in vaginal lactobacilli, which seems to allow the growth of gram-positive bacteria, (especially *Staphylococcus saprophyticus*, *Escherichia coli*, *Enterococcus faecalis*, and *Streptococcus agalactiae*) or *Gardnerella vaginalis*.
	These vaginal bacteria may be present in the vaginal canal and colonize the urinary system, avoiding the immune response and allowing the formation of *E.coli* reservoirs.
Role of urinary microbiome [61,77,141]	Urinary system microbiota has a key role in preserving urinary health. So, the pathophysiology of rUTI is influenced by urobiome.
	Indeed, urinary microbiome composition differs between healthy and rUTIs subjects. Specific urine microorganisms are linked to distinct clinical features in women with rUTI.
Risk factors of rUTIs and dysbiosis [10,76,77,78,79]	Risk factors for the development of symptoms include host variables, host behaviours, and bacterial features. Among these, menopause influences the urine microbiota composition following aging and the decrease in oestreogens protection. First of all, it brings altered Lactobacillus composition, increasing the risk of rUTIs.
Possible immunological pathways [48,66,122,130,139]	Several microscopic pathways have been identified, including the intracellular bacterial community, QIR, LPS, multimicrobial infection, and urothelial mucosal remodelling. These mechanisms allow uropathogens to persist in the bladder and survive antibiotic therapy and host immune response. Furthermore, immunological defences show some abnormalities in UTI-prone women, such as increased levels of IL-12, absence of T-cell response, less VEGF, lower level of monocyte chemotactic protein 1, the upregulation of immediate-early (IE) genes, such Nur77.
New perspectiver of diagnosis [82]	NGS is more sensitive than a conventional urine culture in the detection of uropathogens, highlighting an increased microbiome diversity in the recurrent cystitis group. Additional NGS tests can facilitate rapid decision-making and therapeutic advancement.
New perspectives of theraphy [8,32,38,55,140,144,145,152]	Following the understanding of the importance of lactobacilli and oestrogen in the pathophysiology of rUTIs, several studies demonstrated their benefits as therapies.
	The intravaginal administration of lactobacillus and/or oestrogens is associated with a significant reduction in rUTIs, especially if they are integrated with nonantibiotic therapeutical options as well as modification of behaviour, specific diet, integration with probiotics, and d-mannos, use of local oestrogens therapy, and systemic or local immunostimulants. The administration of one or more of these approaches provides the beneficial treatment to reduce rUTI risk.

## Figures and Tables

**Figure 1 healthcare-11-00525-f001:**
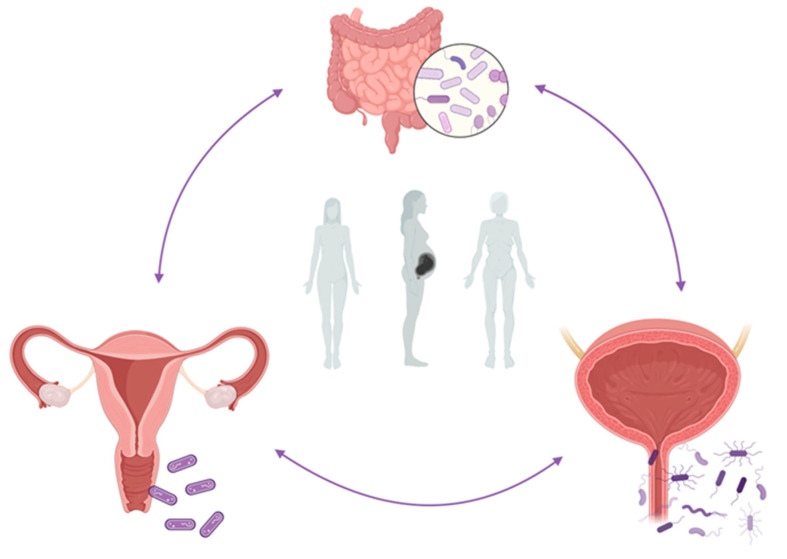
Abstract figure. The interplay between the gut, vaginal, and urinary microbiome in the onset of rUTIs during woman different ages. Figure created with BioRender.com (accessed on 29 January 2023).

**Figure 2 healthcare-11-00525-f002:**
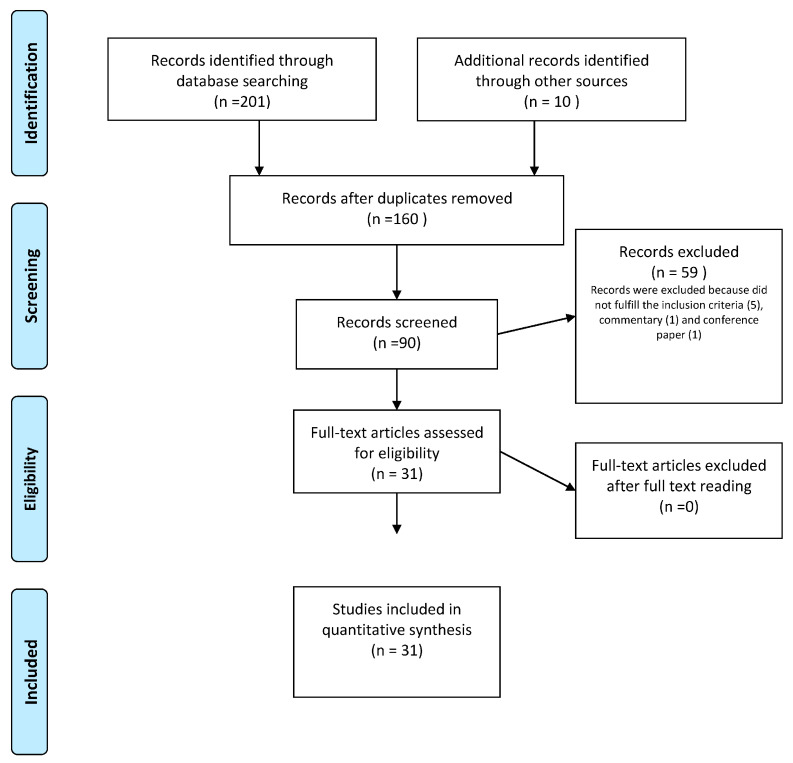
PRISMA flow diagram of study selection process.

## Data Availability

The literature data analyzed during the current review are available from the corresponding author on reasonable request.

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
