# Peer review of "Microbiota Ecosystem in Recurrent Cystitis and the Immunological Microenvironment of Urothelium"

_healthcare, 2023, doi:10.3390/healthcare11040525_

Round 1

Reviewer 1 Report

It is difficult to follow the conclusions of the paper because of the ambiguity of sentences. A thorough proof reading would be beneficial. 

Author Response

Dear reviewer, thank you for your opinion and advice. Following your suggestion, we made a proof reading and an English grammar check was performed by a native speaker.

Reviewer 2 Report

This is a comprehensive review on the recurrence of urinary tract infections, In general, it addresses the problem mainly in adults, however, it would have been interesting to include UTI problems in children with uncomplicated infections. It would be convenient to refer to the criteria to define the differences between recurrent reinfections and persistent reinfections, since this would allow a better understanding of the participation of some pathogens. It is necessary to review some errors in the drafting and writing since some errors were identified.

Author Response

Dear reviewer, thank you for your opinion. Following your advice, we added the criteria to define the differences between recurrent reinfections and persistent reinfections at line 42-43. In addition, an English grammar check was performed by a native speaker.

However, we think that add anything regarding infections in children is beyond the scope of the research: indeed, our aim is to describe how vaginal and urinary microbiome can trigger rUTIs in women, as described in the inclusion criteria. Furthermore, all the authors of this article are gynaecologists.

Reviewer 3 Report

The purpose of this review is to highlight potential mechanisms by which the vaginal and urinary microbiomes in women contribute to rUTIs as well as potential role 59 immunological microenvironment of urothelial in their pathogenies. In fact, the complex correlation among microbiota, low genital tract and urinary system is based on the balance between host characteristics, immunological microenvironment and pathogens.

the paper is well written and illustrated. Bibliographic data is well represented and discussed. After checking for some spelling mistakes, I consider the paper acceptable for publication

Author Response

(The authors gave the same response as above.)

Reviewer 4 Report

The review was conducted well in terms of paper selection according to relevance, and this is a very interesting topic that should be explored more. In this sense, the review brings a good overview of the new perspectives and literature on the topic.

But I have a few suggestions.

1 - Can you add a figure abstract?

2 - Can you please try to decrease the keywords to 3-4?

3 - revision of the English is needed. Can you please review the typos? Below I show some observed in the abstract, but all text would benefit from a check.

For example:  

"most low" is actually lowest (line 15);

 "in female population" should be "in the female population" (line 15); 

"six month" is "six months" (line 17);

 "though" is "thought" (line 18);

"of recurrence" is "for the recurrence" (line 18);

"role of microbiota" is "role of the microbiota" (line 19);

"role of gut" is "role of the gut" (line 20);

"about vaginal and urinary microbiome" is "about the vaginal and urinary microbiomes" (line 21)

4 - I noticed that a few bacteria were not in italic, for example, line 156, Rhodanobacter. Please check it.

 5 - I think the topic of CSTs is not a completely accepted characterization of bacterial vaginosis risk. Mainly because not all people with a profile IV have bacterial vaginosis (no symptoms). Moreover, CST IV are so much more common among Latins and Afro-descendants (10.1371/journal.pone.0181135; 10.7717/peerj.14449). How do you interpret this fact regarding rUTIs? Would you then say that Latins and Afro-descendants would be more prone to rUTIs?

 6 - In the text, you mention reproductive age. And during pregnancy, the microbiome changes considerably towards a lactobacillus-rich microbiome (10.1038/s41591-019-0483-6). Are there any studies regarding pregnancy and rUTIs? Can you add a paragraph on the topic to the review? How prevalent are rUTIs during pregnancy?

 7 - You mention the use of NGS for the diagnostic of rUTIs and say: "NGS is not greatly impacted by antibiotic usage" can you please explain what you mean? Because if the bacteria are long dead by antibiotic treatment, there will be no material to be sequenced. Do you mean the bacteria doesn't need to be alive as for a culture method? If so, can you rephrase it?

8 - Also, in this regard, you mention that sequencing is a new approach to diagnosis. Can you add a paragraph regarding the evolution of sequencing techniques that now allow for the urinary microbiome? Since this is a low biomass sample, it is tricky to sequence and get meaningful data, so an overview could be interesting. 

Author Response

Dear reviewer, thank you for your opinion and advices. 

As you suggested, we added a figure (named Figure 1) created with BioRender.com.

We decreased the keywords to 4: recurrent urinary tract infection, vaginal microbiota, urinary microbiota, dysbiosis.

A proof reading and an English grammar check was performed by a native speaker. All the names of bacteria are in italic now. 

Regarding your consideration at point 5, we agree with you that CSTs are not an exhaustive characterization of bacterial vaginosis risk, but that they are only a good representation and synthesis of the possible vaginal microbiome profiles. Some research, like the two you mentioned above, demonstrated that ethnicity is an independent factor which influences the vaginal microbiota composition, but it is not a risk factor itself for vaginosis and/or rUTIs. As we can read in a lot of papers and what we want this review to convey is that the real risk factor is anything that disturb a previous equilibrium, like the change of Lactobacilli levels.

Following your suggestion, we added a paragraph regarding UTIs and pregnancy at lines 200-211 and regarding evolution of NGS at lines 235-244, and we rephrase the sentence at lines 251-252.

Reviewer 5 Report

Overall the manuscript is quite interesting and useful to mankind. I suggest you to add some natural sources in therapy and prevention part of discussion to treat UTI for eg latest workdone in 2022 on plant bacopa monnieri to treat UTI 

Author Response

Dear reviewer, thank you for your opinion and advice. Following your suggestion, we added some acknowledgments about natural sources of therapy at lines 362-367.

Round 2

Reviewer 4 Report

Thank you for making the changes and accepting my comments.